# Design and Implementation of a New Local Alignment Algorithm for Multilayer Networks

**DOI:** 10.3390/e24091272

**Published:** 2022-09-09

**Authors:** Marianna Milano, Pietro Hiram Guzzi, Mario Cannataro

**Affiliations:** Department of Medical and Surgical Sciences, Data Analytics Research Center, University Magna Græcia, 88100 Catanzaro, Italy

**Keywords:** multilayer network, network alignment, local network alignment

## Abstract

Network alignment (NA) is a popular research field that aims to develop algorithms for comparing networks. Applications of network alignment span many fields, from biology to social network analysis. NA comes in two forms: global network alignment (GNA), which aims to find a global similarity, and LNA, which aims to find local regions of similarity. Recently, there has been an increasing interest in introducing complex network models such as multilayer networks. Multilayer networks are common in many application scenarios, such as modelling of relations among people in a social network or representing the interplay of different molecules in a cell or different cells in the brain. Consequently, the need to introduce algorithms for the comparison of such multilayer networks, i.e., local network alignment, arises. Existing algorithms for LNA do not perform well on multilayer networks since they cannot consider inter-layer edges. Thus, we propose local alignment of multilayer networks (MultiLoAl), a novel algorithm for the local alignment of multilayer networks. We define the local alignment of multilayer networks and propose a heuristic for solving it. We present an extensive assessment indicating the strength of the algorithm. Furthermore, we implemented a synthetic multilayer network generator to build the data for the algorithm’s evaluation.

## 1. Introduction

Network theory is one of the most important frameworks for a meaningful description and an efficient analysis of many complex systems [1,2,3]. Most popular analysis algorithms comprise the mining of a single network, e.g., using community detection algorithms [4]. In parallel, the comparison of networks has led to the introduction of many algorithms for comparing the structures on both a global and a local scale [5,6] that fall into the class of network alignment (NA) algorithms. The first class of algorithms, also known as global network alignment (GNA) algorithms, aims to find the overall similarity among networks. Differently, algorithms belonging to the second group are called local network alignment (LNA) algorithms and aim to find (relatively) small regions of similarity. The output of LNA algorithms is a set of matched regions (or subgraphs) among two graphs given as the input.

More recently, in many application fields, e.g., mobile and social networks and connectomics and metabolomics studies, the need for introducing models more complex than traditional networks arises [7,8]. In such contexts, nodes may have different classes of interactions among them, and such interactions may also be time-varying. In particular, networks representing multiple different associations among patients can be represented by a multilayer graph comprised of multiple interdependent graphs, where each graph represents an aspect or a set of similar interactions [9,10]. Figure 1 represents a simple multilayer graph with three layers. Each layer is a different graph *G*. Edges of a multilayer graph can be *intra-layer*, i.e., connecting nodes of the same layer, and *inter-layer*, i.e., connecting nodes of two different layers [11,12].

Formally, a multilayer network graph may be described as a tuple $G_{ml}$=$V_{L},E_{intra_{L}},$$E_{inter_{L}xL}$, where L={0,1,…,l} is a set of layers and $E_{inter_{L}xL}$ is a set of edges among layers. For each layer *k*, we have a graph $V_{k},E_{intra_{k}}$ (intra-layer edges), and for each pair of layers, *k*, *h* we have a set of edges $Einter_{v}xk$, which is a set of layers connecting nodes of the layers v and k [13].

Examples of multilayer networks come from many different fields, from social network analysis to biological networks. For instance, Figure 2 represents an example of a biological multilayer network representing the interplay among diseases, genes, and drugs.

While many efforts have been made to address challenges related to the analysis of a single network, i.e., community detection in multilayer graphs, there is a need for the formalization and introduction of algorithms to compare multilayer networks. A simple strategy is an adaption or the simple use of existing algorithms for LNA. Unfortunately, this strategy is unsuitable, as previously demonstrated also in heterogeneous networks [14] because the the current algorithms are not able to manage the difference among layers.

Thus, we propose local alignment of multilayer networks (MultiLoAl), a novel algorithm for the local alignment of multilayer networks. We define the local alignment of multilayer networks and propose a heuristic for solving it. MultiLoAl is based on an extension of the previous L-HetNetAligner [14], so it is based on the following steps, as depicted in Figure 3. Our algorithm receives two multilayer networks and a set of similarity relationships among nodes of the same layer in both networks used as the seed to build the alignment.

For instance, considering biological networks, similarity relations are represented by orthologs. The user may find these relations in databases of orthologs (e.g., OrthoMCL, etc.). It produces a set of multilayer graphs representing single local regions of the alignment.

The algorithm merges two input multilayer graphs into a single one, named the multilayer alignment graph, a multilayer graph with the same number of layers of the two inputs, and each layer represents an alignment graph of the same layer of the two input ones. For each node of a layer *k*, the alignment graph features pairs of nodes of the input ones. After building each alignment graph for each layer separately, we analyse the two input graphs to add inter-layer edges of the multilayer alignment graph. Finally, the algorithm uses a community detection algorithm suitable for multilayer graphs to detect communities representing local regions of similarity, i.e., a single region of the alignment. The result of our algorithm is a list of mappings among a subset of nodes of two networks, i.e., a set of mapped regions among input graphs.

We also realized a preliminary implementation of our algorithm by using the R programming language. We here refined such an implementation even in a high performance computing (HPC) fashion and provided deeper experimentation on a larger dataset. The main contributions of this paper are: (i) the implementation of a novel algorithm for the local alignment of multilayer networks, (ii) the definition of the local alignment of multilayer networks, (iii) the solution of a heuristic for solving it, and (iv) the implementation of a synthetic multilayer network generator to build the data for the algorithm evaluation.

The rest of this paper is organized as follows. Section 2 discusses the background on multilayer networks and multilayer community detection. Section 3 presents the MultiLoAl algorithm. Section 4 presents and discusses the results. Finally, Section 5 concludes the paper.

## 2. Related Work

### 2.1. Alignment of Multilayer Networks

The alignment of networks aims to compare two or more networks [5], and existing algorithms may be categorised as local or global based on the approach, despite the existence of other classifications, i.e., algorithms for specific networks such as heterogeneous or temporal networks. Local network alignment (LNA) algorithms aim to find some similar (relatively) small subnetworks, while global network alignment (GNAs) algorithms search for the best superimposition of the whole compared networks. The literature contains many algorithms for other classes of networks (see, for instance, [5]). Unfortunately, the existing algorithms do not perform very well for multilayer networks [11,15].

### 2.2. Community Detection in Multilayer Networks

Community detection is one of the most popular research areas in various complex systems, such as biology, sociology, medicine, and transportation systems [16,17,18]. The reason is that the community structures, defined as groups of nodes that are more densely connected than the rest of the network, represent significant characteristics for understanding the functionalities and organisations of complex systems modelled as networks [16]. It is expected that the communities play significant roles in the structure–function relationship. For example, in biological networks such as protein–protein interaction (PPI) networks, the communities represent proteins involved in a similar function; in neuroscience, the communities detected in brain networks mean regions of interest (ROI) that are active during tasks; in social networks, communities can be groups of friends or colleagues; in the World Wild Web, the communities represent the web pages sharing the same topic [19]. Thus, the discovery of communities in these systems has become an interesting approach to figuring out how network structure relates to system behaviours.

Discovering a community structure in multilayer networks has became a hot research topic due to the inability of classical community detection methods to deal with the complexity of the multilayer model.

In fact, in multilayer networks, the communities represent groups of well-connected nodes in all layers. Thus, the detection algorithms should take into account the differences among layers. Unfortunately, traditional community detection methods are not able to deal with the complexity of the multilayer networks because (i) they do not enable analysing subsets of the layers and also (ii) they do not depict the diverse layers, and thus, they cannot distinguish between different types of multilayer communities [20]. To overcome these limitations, different community detection algorithms for multilayer networks have been recently proposed. For example, Infomap [21], a multilayer generalization of [22], is a method based on random walks. It considers that an entity randomly following the edges of a network would tend to become captured within communities due to the greater density of edges between nodes within the same community, moving less frequently from one community to another. This algorithm tries to identify a partition of vertices and levels that minimises the equation of the generalised map, which measures the length of the description of a random walk on the partition.

GenLouvain [23] is a multilayer generalisation of the iterative GenLouvain algorithm. This algorithm seeks a partition of the nodes and layers that maximises the multilayer modularity of the network. It searches the global information of the network, finding which are the edges of the network that contribute to the creation of the community structure; then, it applies a novel measure of edge centrality, to classify all the edges of the network concerning their proclivity to propagate information through the network itself.

ABACUS [24] is an algorithm that ensures the mining of multidimensional communities based on the extraction of frequent closed itemsets from monodimensional community memberships. At first, ABACUS considers each dimension independently, and it mines monodimensional communities. After that, it labels each node with a list of pair tags, i.e., the dimension and community the node belongs to in that dimension. Then, ABACUS considers each pair of tags as an item, and it applies a frequent-closed-itemset-mining algorithm. Finally, the multidimensional communities described by the itemsets consist of frequent closed itemsets.

Multilayer clique percolation [25] is a method that extends the popular clique percolation method for simple networks, where dense regions correspond to cliques and adjacency consists of having common nodes. The algorithm extends this step by searching cliques by encompassing multiple layers and reformulating adjacency so that both common nodes and common layers are expected. Multilayer clique percolation communities are combinations of adjacent cliques, so all the edges in these cliques can be considered part of the community.

Multidimensional label propagation (mdlp) [26] is an algorithm based on label propagation. At first, the algorithm assigns a different label to each node, and then, it weights the contribution of each neighbour based on their similarity with the nodes on the different layers. In particular, if two nodes are adjacent on all layers and have the same neighbours, they would have a higher probability of sharing the same label. Finally, the algorithm gives a score for each pair of nodes, referring to how likely a label should be extended from one to the other, bringing a common community.

## 3. MultiLoAl Algorithm

Initially, the algorithm inputs two multilayer networks and a set of similarities among node pairs of the same layer into the input networks. Then, it builds the alignment by performing two main steps: (i) construction of the multilayer alignment graph and (ii) mining of the multilayer alignment graph.

MultiLoAl analyses separately each corresponding pair of the corresponding layers of the input graphs. Each pair of a network of the same layer builds an alignment graph, as previously shown in L-HetNetAligner [14]. Then, it analyses the inter-layer edges of the input networks to add inter-layer edges to the multilayer alignment graph. Once the alignment graph is built, we use an algorithm for detecting communities in multilayer networks to uncover relevant modules. Figure 4 shows these steps.

MultiLoAl is a novel algorithm for the local alignment of multilayer networks. MultiLoAl builds the alignment on two main steps, as depicted in Figure 3:(i) Construction of the multilayer alignment graph;(ii) Analysis of the alignment graph.

Step (i) may be subdivided into two substeps: (i.a) adding nodes and intra-layer edges; (i.b) adding inter-layer edges.

Let us consider two multilayer input graphs $G_{1}$ and $G_{2}$.

Node colours are used to distinguish different types of nodes belonging to two different types of layers. For simplicity, two multilayer input networks have the same number of nodes.

### 3.1. Step (1.a): Adding Nodes and Intra-Layer Edges to the Alignment Graph

In the first step, the algorithm considers each pair of corresponding layers separately see Figure 5. For each layer, it builds an alignment graph following the approach proposed in L-HetNetAligner [14], adapted to the case of one-colour networks, as reported in Figure 6.

At this stage, the algorithm, starting from an initial list of seed nodes, builds the alignment graph by initially constructing two intermediate alignment graphs, which we call alignment graph layer 1 and alignment graph layer 2, for two networks belonging to layer 1 and two networks belonging to layer 2. Therefore, we define the alignment graph $G{al}_{=}\left(V_{al},E_{al}\right)$ as a graph constructed by two initial input graphs $G_{1}=\left(V_{1},E_{1}\right)$ and $G_{2}=\left(V_{2},E_{2}\right)$. Each node $v_{al}\in V_{al}$ represents the matching of nodes of the input graphs, so $V_{al}\subseteq V_{1}\times V_{2}$. The selection of node pairs is guided by the input similarity relationships. Therefore, each node is matched with the most similar node of the other network through the use of the input similarity relationships, i.e., seed nodes; each node of the alignment graph represents a pair of similarities among nodes from the input networks; see Figure 7.

Once all nodes have been added to the graph, the algorithm builds the edges of the alignment graphs. For each pair of nodes, the algorithm examines the two input graphs, and it inserts and weights the edges considering three conditions: match, mismatch, and gap. Let us consider the nodes of the alignment graphs; in particular, let us consider the pair of nodes (G1−G1) and (G2−G2) in Figure 6. To determine the presence of an edge, we consider the edge $\left(G1,G2\right)\in G_{1}$ network and $\left(G1,G2\right)\in G_{2}$ network. If $G_{1}$ and $G_{2}$ contain these nodes and the nodes are adjacent, there is a **match**, which we call, for convenience, a **homogeneous match**, since the nodes of the two networks are of the same type (see Figure 8a).

Let us consider Δ=2 as the node distance, i.e., the length of the shortest connecting path threshold to discriminate between gaps and mismatches. If $G_{1}$ and $G_{2}$ contain these nodes and the nodes are adjacent only in a single network, there is a **mismatch**, which we call a **homogeneous mismatch** (Figure 8b).

If $G_{1}$ and $G_{2}$ contain these nodes, the nodes are adjacent only in a single network, and they are at a distance less than Δ (gap threshold) in the other network, there is a **gap**, which we call a **homogeneous gap** (Figure 8c). After the edges of the alignment, graphs are added, and a weight is assigned to each edge by applying an ad hoc scoring function *F* and the gap threshold Δ. The function assigns a high score to the matches than to the mismatches and gaps. The kind of scoring function has a large significance for the resulting alignment graph and on the alignment itself. The algorithm enables the user to choose other values to optimize the quality of the results. In this work, we set the weight assigned to each edge as follows: homogeneous match equal to 1, homogeneous mismatch equal to 0.5, homogeneous gap equal to 0.2.

### 3.2. Step (1.b): Adding Inter-Layer Edges

The algorithm adds the inter-layer edges among multilayer alignment graph layer 1 and alignment graph layer 2. For each pair of nodes in the multilayer alignment graphs, the algorithm examines the corresponding layers of the input graphs. Let us consider the pair of nodes (G1) and (D4) in Figure 8. To determine the presence of an edge, we consider the edge $\left(G1,D4\right)\in G_{1}$ network and $\left(G1,D4\right)\in G_{2}$ network. The initial graph contains both edges connecting their internal nodes, and if the nodes are adjacent, there is a **match**, which we call, for convenience, a **heterogeneous match**, since the nodes of the two networks are of different types; see Figure 9a.

Let us consider the pair of nodes (G5) and (D2) in Figure 8b. To determine the presence of an edge, we consider the edge $\left(G5,D2\right)\in G_{1}$ network and $\left(G5,D2\right)\in G_{2}$ network. $G_{1}$ contains the edge (G5,D2), while nodes G5 and D2 are disconnected in $G_{2}$ If the initial graph contains both edges connecting their internal nodes and the nodes are adjacent, there is a **match**, which we call, for convenience, a **heterogeneous match**, since the nodes of the two networks are of different types; see Figure 9a. Therefore, there is a **heterogeneous mismatch** (Figure 9b). Then, we set the weight assigned to each edge as follows: heterogeneous match equal to 0.9, heterogeneous mismatch equal to 0.4.

### 3.3. Step 2: Detection of Communities on the Alignment Graph

Finally, the final alignment graph is then mined to discover communities by applying a community detection algorithm by using existing algorithms for multilayer networks [27,28,29,30], see Figure 10. Our methodology presents a general design, so it is possible to mine the final alignment graph by applying a different mining method.

In the current version of MultiLoAl, we applied the Infomap algorithm to mine the communities on the alignment graph. However, the user can choose which community detection algorithm to select among Generalized Louvain, ABACUS, clique percolation, and mdlp. The output consists of a file that contains the extracted communities as a list of nodes, the weight of the edge, and the string in which it is reported if there is a homogeneous/heterogeneous match, homogeneous/mismatch, or homogeneous gap (see an example of the output at https://github.com/mmilano87/MultiLoAl (accessed on 12 August 2022)).

### 3.4. MultiLoAl vs. L-HetNetAligner

MultiLoAl, despite being based on the previous L-HetNetAligner, presents many different characteristics. First, the algorithms have different scopes: MultiLoAl is a local aligner of multilayer networks, while L-HetNetAligner works only on heterogeneous networks. In detail, by analysing the building of local alignment, MultiLoAl and L-HetNetAligner have two main general steps: (i) construction of the alignment graph; (ii) mining of the alignment graph. The building of the alignment graph is the first main difference among the two algorithms. In fact, MultiLoAl builds a multilayer alignment graph through two substeps: (i) by adding nodes and intra-layer edges, following the approach proposed in L-HetNetAligner adapted to the case of one-colour networks; (ii) by adding inter-layer edges. This last step represents the main novelty compared to L-HetNetAligner, because MultiLoAl analyses and adds the edges among different layers of input networks. Otherwise, L-HetNetAligner builds a heterogeneous alignment graph. Initially, L-HetNetAligner defines the nodes of the alignment graph as composite nodes representing pairs of nodes matched by the similarity considerations. The algorithm inserts and weights the edges in the alignment graph to the nodes for which the edge links have the same colour and according to their distance in the input network. Finally, once the alignment graph is built, both algorithms mine the alignment graph to discover modules that represent local alignment. MultiLoAl applies a community detection algorithm, Infomap, to mine the final alignment. The result consists of the extracted communities as a list of nodes, the weight of the edge, and the string in which it is reported if there is a homogeneous/heterogeneous match, homogeneous/mismatch, or homogeneous gap. Otherwise, L-HetNetAligner uses the Markov clustering (MCL) algorithm to cluster the graph. Each extracted module represents a single region of the alignment. The result of our algorithm is a list of mappings among a subset of nodes of two networks, i.e., a set of mapped regions among input graphs.

## 4. Results and Discussion

### 4.1. Evaluation of the Quality of the Alignment

The evaluation of the quality of the alignment of network is still a matter of debate for simple networks [5,31,32]. There exist many measures able to evaluate both the correctness of the alignment, as well as the quality of the obtained alignment [33]. On the other side, there is no gold standard to benchmark the alignment. Moreover, all the existing measures need to be extended in the multilayer case. Thus, we first introduce novel measures of correctness in the multilayer case (to the best of our knowledge, there are not any other available measures), then we perform an assessment of our methods. We first designed a proof of concept to show the ability of our algorithm to map correct nodes and edges by aligning a synthetic network with itself and with some randomised versions.

The correctness of an alignment is usually evaluated by means of the analysis of its topological quality, i.e., the ability to reconstruct the underlying true node mapping well (when such a mapping is known) and if it conserves many edges. For simple networks, F−NC (F-score node correctness) is a measure of node correctness, and it is a combination of two measures: P−NC and R−NC. P−NC is calculated as $\frac{M\cap N}{M}$, and R−NC is defined as $\frac{M\cap N}{N}$, where *M* is the set of node pairs that are mapped under the true node mapping and *N* is the set of node pairs that are aligned under an alignment *f*.

We here extended such a measure in the multilayer case. We first considered in a separate way each layer, and we calculated the $F-{\mathrm{NC}}_{\rangle }$ for each layer 〉. Then, we computed the *multilayer*$F-{\mathrm{NC}}_{⇕}$ as the average of all $F-{\mathrm{NC}}_{\rangle }$.

Similarly, the edge correctness in the simple case can be measured by considering *NCV-G$S^{3}$*, which is a combination of two measures: high node coverage (NCV) and generalized $S^{3}$ (G$S^{3}$). NCV is the percentage of nodes from $G_{1}$ and $G_{2}$ that are also in $G_{1}^{\prime }$ and $G_{2}^{\prime }$, and G$S^{3}$ measures how well edges are conserved between $G_{1}^{\prime }$ and $G_{2}^{\prime }$, where $G_{1}$ and $G_{2}$ are two graphs and $G_{1}^{\prime }$ and $G_{2}^{\prime }$ are subgraphs of $G_{1}$ and $G_{2}$ that are induced by the mapping.

We used NCV-G$S^{3}$ to measure the edge correctness of each layer 〉, then we averaged the measures of such values for all the layers, and we obtained the multilayer NCV-G$S^{3}$.

Finally, we should consider the edge correctness for the inter-layer edges. Without loss of information, we considered all the inter-layer edges as a whole, and we calculated the correctness of all the inter-layer edges as $\mathrm{NCV}-{\mathrm{GS}}_{\rangle }^{∋}\backslash ⊔\rceil \nabla$.

### 4.2. Proof of Concept

As a proof of principle, we present the use of the MultiLoAl dataset consisting of ten multilayer synthetic networks that we built with the graph generator, implemented ad hoc in the R code. An example of the multilayer network and R function are available on the web site of the project (https://github.com/mmilano87/MultiLoAl (accessed on 12 August 2022)).

All the multilayer networks have 30 nodes and 2 layers, whereas the edges are distributed as depicted in Table 1.

First, we aligned each network with respect to itself to show the ability to find known regions of similarity; second, we considered the alignment of the network with respect to an altered version of the network obtained by adding different levels of noise (5%, 10%, 15%, 20%, and 25%) by randomly removing edges from the network. The aim of the test was to demonstrate that the alignment algorithms are capable of producing high-quality alignments with an edge conservation of about 90%. Then, we implemented different versions of the MultiLoAl algorithm by varying the strategy applied to mine the community on the alignment graph. We executed the experiments on an Intel Core i5 Processor, 2.9 Ghz, with 4 Gbytes of main memory running the Ubuntu OS ver 18.04. MultiLoAl built 60 alignments, and it completed the whole process of alignment in ten seconds.

To measure the performance of the alignments built with different versions of MultiLoAl, we evaluated the quality of the results by considering the topological aspects of alignments and the number of communities found. At first, the results were evaluated by the topological quality.

We computed the NCV-G$S^{3}$ and F-NC measures for all alignment networks by considering the intra-layer and inter-layer edges. Table 2, Table 3, Table 4 and Table 5 report the results. Table 6, Table 7, Table 8 and Table 9 report the mean and standard deviation values of the NCV-G$S^{3}$ and F-NC measures for each synthetic network aligned with its noisy counterpart.

The results show that the quality of the alignment was greater when Infomap was applied to mine the community. Furthermore, increasing the noise level from 5% to 25% in the original networks caused NCV-G$S^{3}$ and F-NC to decrease.

## 5. Conclusions

Recently, the applications of multilayer networks in social network analysis, in finance, and in biology have been increasing. Multilayer networks can be seen as a set of networks (each network is a distinct layer) connected by inter-layer links. We here focused on the problem of comparing two multilayer networks, highlighting small local regions of similarity. Since existing algorithms for simple networks do not perform well on multilayer networks, we proposed Local Alignment of Multilayer Networks (MultiLoAl), a novel algorithm for the local alignment of multilayer networks. We proposed a heuristic for solving it. Furthermore, we performed an extensive evaluation to reveal the strength of the algorithm. Since we presented the use of MultiLoAl on multilayer synthetic networks, we plan to extend the application to real biological networks.

## Figures and Tables

**Figure 1 entropy-24-01272-f001:**
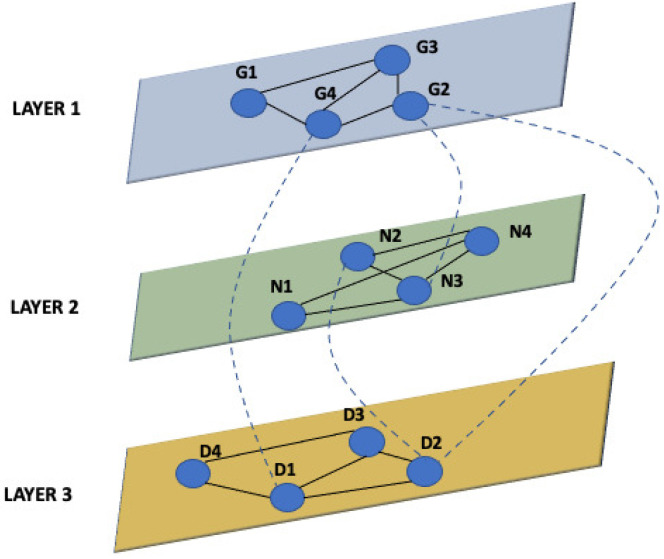
Example of a multilayer network. The figure represents a simple multilayer graph with three layers. Each layer is a different graph. Edges of a multilayer graph can be *intra-layer*, i.e., connecting nodes of the same layer, and *inter-layer*, i.e., connecting nodes of two different layers.

**Figure 2 entropy-24-01272-f002:**
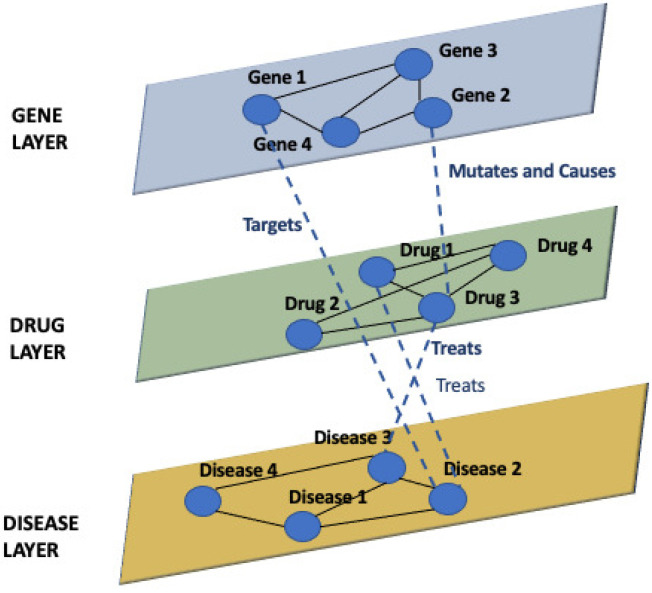
Example of a biological multilayer network.

**Figure 3 entropy-24-01272-f003:**
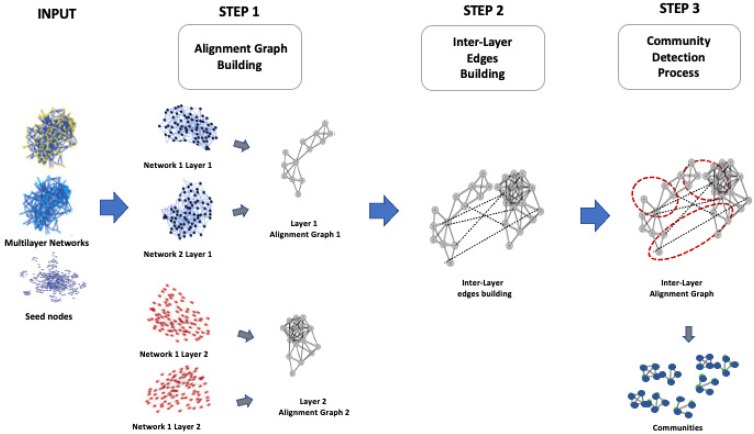
Local alignment of multilayer networks.

**Figure 4 entropy-24-01272-f004:**
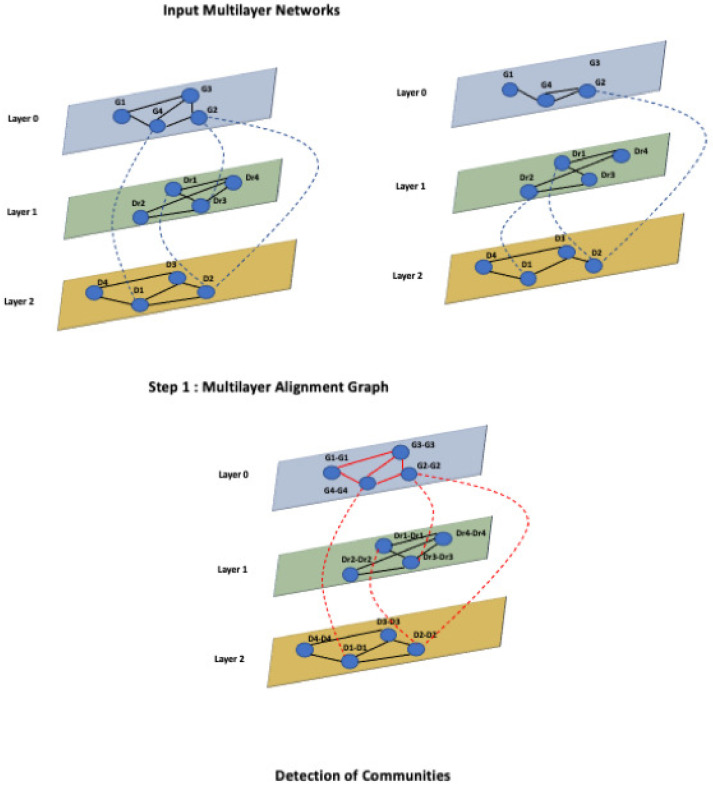
Workflow of the proposed algorithm.

**Figure 5 entropy-24-01272-f005:**
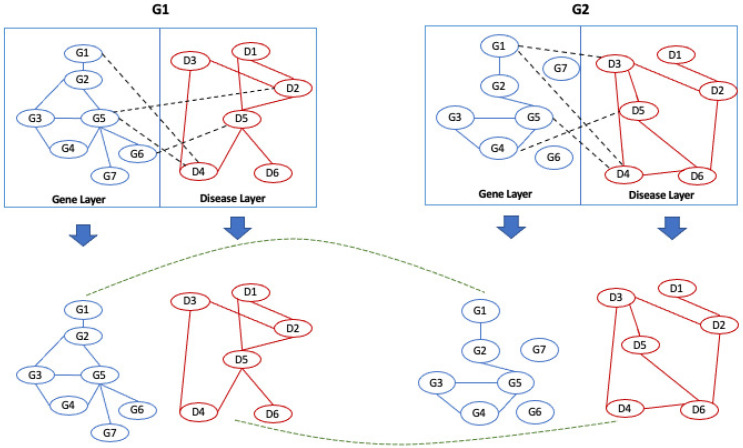
The algorithm separates the input networks according to the layer type.

**Figure 6 entropy-24-01272-f006:**
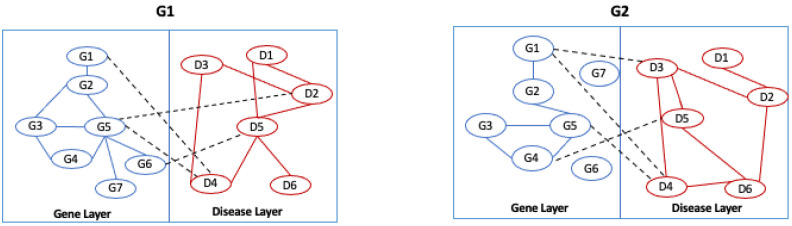
Example of a multilayer network. The nodes represents a set of genes and a set of diseases that belong to the gene layer and the disease layer.

**Figure 7 entropy-24-01272-f007:**
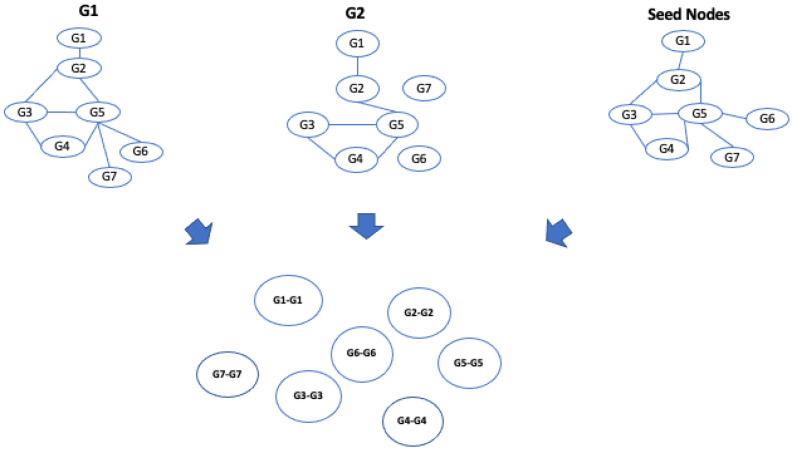
Building of the alignment graph: node definition. The algorithm takes the two networks and a subset of node pairs matched according to a similarity function and starts to build the alignment graph. In this step, the algorithm defines the nodes of the alignment graph represented by the pair of matched nodes.

**Figure 8 entropy-24-01272-f008:**
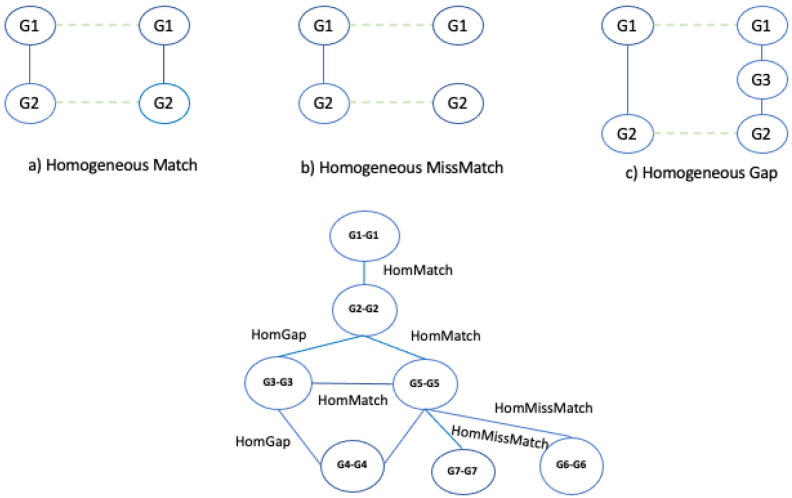
Example of homogeneous match, homogeneous mismatch, and homogeneous gap and building of alignment graph.

**Figure 9 entropy-24-01272-f009:**
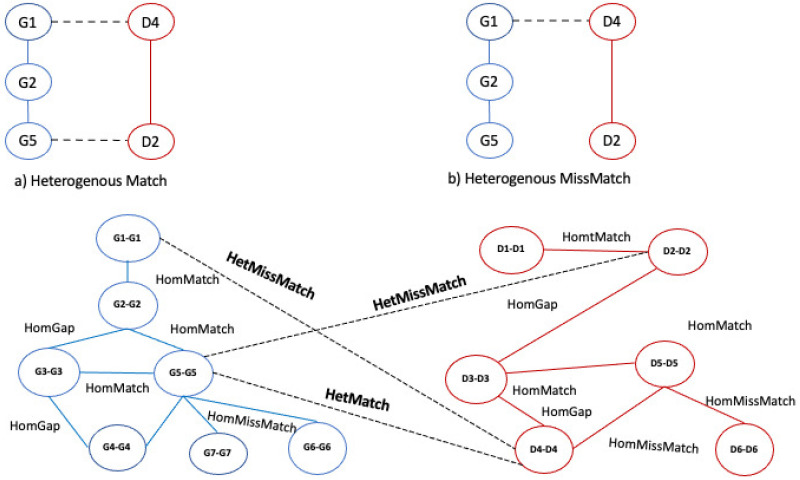
Example of a homogeneous match, mismatch, and gap and building of inter-layer alignment graph.

**Figure 10 entropy-24-01272-f010:**
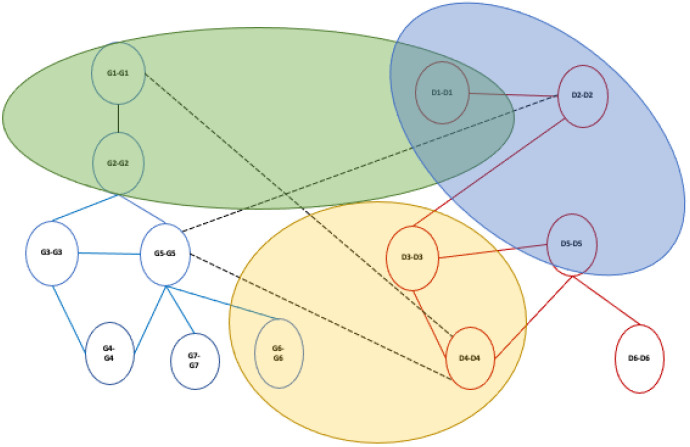
Example of community detection extraction on inter-layer alignment graph.

**Table 1 entropy-24-01272-t001:** Characteristics of the synthetic multilayer networks.

Network	Layers	Nodes	Edges
N1	2	30	90
N2	2	30	96
N3	2	30	84
N4	2	30	78
N5	2	30	95
N6	2	30	88
N7	2	30	93
N8	2	30	83
N9	2	30	94
N10	2	30	96

**Table 2 entropy-24-01272-t002:** NCV-G$S^{3}$ values computed on intra-layer edges for all the alignments by applying Infomap, Generalized Louvain, ABACUS, clique percolation, and mdlp community detection algorithms.

Network	Noise	NCV-G$S^{3}$ with Infomap	NCV-G$S^{3}$ with Generalized Louvain	NCV-G$S^{3}$ with ABACUS	NCV-G$S^{3}$ with Clique Percolation	NCV-G$S^{3}$ with mdlp
N1	0	0.99	0.999	0.975	0.966	0.955
	5	0.976	0.968	0.947	0.966	0.954
	10	0.96	0.92	0.925	0.942	0.95
	15	0.926	0.907	0.918	0.887	0.911
	20	0.914	0.891	0.903	0.887	0.906
	25	0.903	0.879	0.901	0.877	0.896
N2	0	0.995	0.996	0.951	0.966	0.976
	5	0.964	0.992	0.843	0.912	0.829
	10	0.943	0.98	0.788	0.872	0.783
	15	0.941	0.953	0.773	0.862	0.708
	20	0.89	0.945	0.767	0.802	0.666
	25	0.881	0.894	0.766	0.792	0.615
N3	0	0.995	0.989	0.977	0.914	0.845
	5	0.972	0.982	0.952	0.881	0.839
	10	0.961	0.956	0.883	0.815	0.836
	15	0.937	0.949	0.855	0.814	0.771
	20	0.933	0.923	0.801	0.73	0.754
	25	0.929	0.881	0.782	0.726	0.657
N4	0	0.979	0.999	0.976	0.854	0.809
	5	0.979	0.97	0.959	0.849	0.744
	10	0.962	0.938	0.892	0.827	0.742
	15	0.915	0.914	0.844	0.82	0.698
	20	0.887	0.886	0.797	0.815	0.694
	25	0.881	0.885	0.788	0.721	0.691
N5	0	0.99	0.992	0.932	0.921	0.839
	5	0.968	0.97	0.921	0.896	0.822
	10	0.957	0.963	0.881	0.89	0.756
	15	0.952	0.937	0.866	0.881	0.718
	20	0.935	0.92	0.845	0.817	0.717
	25	0.901	0.903	0.795	0.741	0.683
N6	0	0.994	0.966	0.969	0.872	0.968
	5	0.968	0.943	0.968	0.81	0.913
	10	0.942	0.941	0.938	0.796	0.884
	15	0.932	0.936	0.836	0.765	0.834
	20	0.916	0.917	0.835	0.751	0.808
	25	0.891	0.909	0.76	0.741	0.665
N7	0	0.966	0.989	0.938	0.86	0.98
	5	0.966	0.987	0.932	0.816	0.978
	10	0.956	0.959	0.855	0.807	0.9
	15	0.943	0.942	0.846	0.795	0.892
	20	0.941	0.898	0.841	0.765	0.87
	25	0.887	0.897	0.84	0.728	0.683
N8	0	0.997	0.96	0.969	0.954	0.969
	5	0.98	0.957	0.955	0.839	0.812
	10	0.966	0.953	0.892	0.836	0.803
	15	0.956	0.951	0.837	0.825	0.791
	20	0.953	0.943	0.782	0.79	0.765
	25	0.922	0.895	0.772	0.726	0.764
N9	0	0.997	0.987	0.97	0.956	0.95
	5	0.936	0.967	0.969	0.937	0.916
	10	0.923	0.967	0.95	0.93	0.786
	15	0.909	0.934	0.915	0.923	0.726
	20	0.882	0.928	0.8	0.874	0.635
	25	0.879	0.905	0.767	0.743	0.616
N10	0	0.991	0.986	0.948	0.972	0.839
	5	0.982	0.958	0.872	0.959	0.825
	10	0.944	0.957	0.853	0.936	0.75
	15	0.938	0.943	0.839	0.871	0.689
	20	0.905	0.938	0.838	0.843	0.623
	25	0.904	0.911	0.811	0.757	0.603

**Table 3 entropy-24-01272-t003:** NCV-G$S^{3}$ values computed on inter-layer edges for all the alignments by applying Infomap, Generalized Louvain, ABACUS, clique percolation, and mdlp community detection algorithms.

Network	Noise	NCV-G$S^{3}$ with Infomap	NCV-G$S^{3}$ with Generalized Louvain	NCV-G$S^{3}$ with ABACUS	NCV-G$S^{3}$ with Clique Percolation	NCV-G$S^{3}$ with mdlp
N1	0	0.76	0.76	0.745	0.711	0.576
	5	0.727	0.727	0.691	0.706	0.571
	10	0.68	0.68	0.676	0.701	0.551
	15	0.641	0.641	0.664	0.678	0.537
	20	0.624	0.624	0.598	0.644	0.513
	25	0.617	0.617	0.567	0.599	0.506
N2	0	0.757	0.782	0.725	0.733	0.579
	5	0.716	0.728	0.72	0.665	0.578
	10	0.709	0.711	0.714	0.662	0.555
	15	0.686	0.639	0.705	0.613	0.551
	20	0.672	0.63	0.701	0.598	0.505
	25	0.639	0.596	0.595	0.553	0.492
N3	0	0.735	0.771	0.727	0.707	0.6
	5	0.734	0.693	0.725	0.705	0.592
	10	0.717	0.662	0.627	0.678	0.592
	15	0.676	0.656	0.61	0.638	0.532
	20	0.663	0.602	0.594	0.611	0.516
	25	0.618	0.589	0.578	0.595	0.504
N4	0	0.666	0.788	0.652	0.71	0.591
	5	0.654	0.679	0.642	0.703	0.579
	10	0.645	0.588	0.632	0.683	0.528
	15	0.643	0.587	0.616	0.652	0.527
	20	0.638	0.586	0.6	0.583	0.509
	25	0.605	0.584	0.588	0.576	0.497
N5	0	0.791	0.781	0.713	0.711	0.591
	5	0.759	0.778	0.703	0.711	0.587
	10	0.721	0.664	0.67	0.666	0.57
	15	0.721	0.607	0.643	0.626	0.529
	20	0.636	0.588	0.634	0.588	0.524
	25	0.636	0.585	0.573	0.564	0.518
N6	0	0.799	0.76	0.685	0.739	0.596
	5	0.794	0.754	0.671	0.708	0.558
	10	0.79	0.729	0.605	0.698	0.538
	15	0.728	0.685	0.604	0.691	0.53
	20	0.719	0.651	0.571	0.552	0.518
	25	0.659	0.636	0.558	0.551	0.51
N7	0	0.754	0.73	0.703	0.629	0.59
	5	0.728	0.697	0.675	0.613	0.571
	10	0.712	0.681	0.654	0.607	0.565
	15	0.706	0.651	0.631	0.57	0.56
	20	0.674	0.617	0.572	0.565	0.502
	25	0.639	0.592	0.563	0.56	0.499
N8	0	0.789	0.783	0.738	0.722	0.584
	5	0.717	0.659	0.666	0.685	0.548
	10	0.684	0.623	0.589	0.639	0.54
	15	0.666	0.622	0.583	0.63	0.515
	20	0.618	0.621	0.576	0.627	0.509
	25	0.615	0.586	0.567	0.62	0.505
N9	0	0.767	0.77	0.727	0.728	0.588
	5	0.75	0.749	0.722	0.674	0.584
	10	0.697	0.748	0.603	0.646	0.574
	15	0.67	0.729	0.575	0.637	0.547
	20	0.662	0.669	0.562	0.616	0.518
	25	0.609	0.584	0.553	0.564	0.504
N10	0	0.783	0.775	0.745	0.737	0.597
	5	0.744	0.761	0.678	0.725	0.586
	10	0.721	0.741	0.666	0.654	0.574
	15	0.703	0.666	0.664	0.653	0.541
	20	0.649	0.654	0.607	0.59	0.528
	25	0.643	0.648	0.598	0.57	0.508

**Table 4 entropy-24-01272-t004:** F-NC values computed on intra-layer edges for all the alignments by applying Infomap, Generalized Louvain, ABACUS, clique percolation, and mdlp community detection algorithms.

Network	Noise	F-NC with Infomap	F-NC with Generalized Louvain	F-NC with ABACUS	F-NC with Clique Percolation	F-NC with mdlp
N1	0	0.626	0.575	0.591	0.53	0.568
	5	0.62	0.566	0.573	0.518	0.54
	10	0.608	0.556	0.558	0.511	0.528
	15	0.601	0.549	0.528	0.502	0.512
	20	0.599	0.528	0.505	0.501	0.504
	25	0.568	0.524	0.501	0.492	0.479
N2	0	0.643	0.605	0.571	0.555	0.552
	5	0.643	0.603	0.566	0.55	0.506
	10	0.619	0.598	0.565	0.537	0.5
	15	0.61	0.596	0.542	0.528	0.483
	20	0.605	0.581	0.534	0.521	0.461
	25	0.587	0.523	0.53	0.51	0.452
N3	0	0.648	0.609	0.58	0.576	0.546
	5	0.609	0.603	0.542	0.556	0.532
	10	0.599	0.598	0.541	0.553	0.532
	15	0.58	0.592	0.516	0.516	0.502
	20	0.578	0.59	0.516	0.516	0.473
	25	0.577	0.57	0.507	0.507	0.47
N4	0	0.628	0.608	0.584	0.576	0.552
	5	0.626	0.592	0.567	0.573	0.54
	10	0.612	0.581	0.558	0.553	0.537
	15	0.57	0.557	0.557	0.544	0.528
	20	0.565	0.53	0.497	0.536	0.509
	25	0.559	0.517	0.495	0.521	0.473
N5	0	0.649	0.601	0.6	0.559	0.531
	5	0.648	0.557	0.592	0.526	0.516
	10	0.637	0.555	0.578	0.483	0.511
	15	0.635	0.542	0.55	0.482	0.485
	20	0.626	0.541	0.547	0.472	0.474
	25	0.618	0.539	0.503	0.471	0.451
N6	0	0.636	0.578	0.587	0.566	0.525
	5	0.633	0.577	0.572	0.56	0.516
	10	0.587	0.546	0.539	0.553	0.499
	15	0.58	0.532	0.527	0.545	0.482
	20	0.568	0.528	0.519	0.507	0.468
	25	0.561	0.514	0.49	0.483	0.467
N7	0	0.642	0.608	0.572	0.567	0.542
	5	0.634	0.606	0.568	0.547	0.537
	10	0.597	0.573	0.543	0.502	0.536
	15	0.597	0.568	0.521	0.491	0.507
	20	0.565	0.511	0.505	0.49	0.501
	25	0.554	0.51	0.499	0.488	0.47
N8	0	0.625	0.59	0.585	0.571	0.551
	5	0.601	0.555	0.565	0.504	0.537
	10	0.591	0.554	0.547	0.492	0.515
	15	0.583	0.541	0.545	0.486	0.514
	20	0.552	0.522	0.54	0.483	0.508
	25	0.551	0.52	0.533	0.472	0.466
N9	0	0.639	0.605	0.594	0.568	0.563
	5	0.632	0.596	0.583	0.56	0.523
	10	0.607	0.574	0.579	0.554	0.513
	15	0.582	0.561	0.555	0.53	0.498
	20	0.575	0.542	0.54	0.512	0.456
	25	0.574	0.523	0.507	0.509	0.455
N10	0	0.641	0.597	0.581	0.57	0.505
	5	0.614	0.579	0.554	0.52	0.492
	10	0.603	0.558	0.522	0.51	0.478
	15	0.557	0.526	0.519	0.494	0.47
	20	0.552	0.526	0.509	0.491	0.469
	25	0.551	0.513	0.491	0.484	0.461

**Table 5 entropy-24-01272-t005:** F-NC values computed on inter-layer edges for all the alignments by applying Infomap, Generalized Louvain, ABACUS, clique percolation, and mdlp community detection algorithms.

Network	Noise	F-NC with Infomap	F-NC with Generalized Louvain	F-NC with ABACUS	F-NC with Clique Percolation	F-NC with mdlp
N1	0	0.566	0.577	0.555	0.548	0.465
	5	0.55	0.573	0.509	0.543	0.462
	10	0.545	0.539	0.496	0.533	0.46
	15	0.533	0.519	0.474	0.493	0.448
	20	0.531	0.516	0.472	0.473	0.441
	25	0.501	0.482	0.467	0.471	0.429
N2	0	0.591	0.566	0.548	0.539	0.485
	5	0.568	0.545	0.548	0.498	0.478
	10	0.522	0.532	0.524	0.497	0.461
	15	0.512	0.517	0.504	0.477	0.452
	20	0.506	0.506	0.483	0.471	0.402
	25	0.503	0.499	0.471	0.455	0.401
N3	0	0.561	0.576	0.553	0.548	0.476
	5	0.559	0.548	0.511	0.548	0.469
	10	0.553	0.547	0.505	0.511	0.447
	15	0.55	0.543	0.489	0.496	0.446
	20	0.531	0.496	0.483	0.467	0.425
	25	0.502	0.485	0.483	0.463	0.425
N4	0	0.596	0.577	0.534	0.543	0.467
	5	0.561	0.574	0.529	0.516	0.465
	10	0.547	0.568	0.513	0.51	0.446
	15	0.53	0.531	0.511	0.49	0.442
	20	0.528	0.505	0.505	0.482	0.44
	25	0.512	0.482	0.462	0.458	0.402
N5	0	0.588	0.569	0.538	0.52	0.47
	5	0.585	0.524	0.527	0.52	0.45
	10	0.585	0.5	0.524	0.512	0.412
	15	0.565	0.486	0.499	0.472	0.403
	20	0.516	0.485	0.492	0.463	0.403
	25	0.507	0.485	0.475	0.455	0.401
N6	0	0.578	0.564	0.543	0.539	0.481
	5	0.565	0.554	0.541	0.533	0.471
	10	0.554	0.542	0.536	0.498	0.466
	15	0.552	0.539	0.525	0.493	0.457
	20	0.536	0.533	0.513	0.492	0.412
	25	0.518	0.518	0.496	0.481	0.41
N7	0	0.587	0.525	0.545	0.548	0.488
	5	0.573	0.513	0.502	0.522	0.479
	10	0.555	0.51	0.495	0.52	0.476
	15	0.529	0.507	0.483	0.514	0.465
	20	0.529	0.496	0.48	0.491	0.462
	25	0.508	0.482	0.471	0.454	0.419
N8	0	0.598	0.579	0.553	0.52	0.486
	5	0.59	0.563	0.532	0.511	0.473
	10	0.578	0.563	0.517	0.494	0.439
	15	0.56	0.561	0.512	0.481	0.437
	20	0.535	0.549	0.478	0.48	0.413
	25	0.524	0.48	0.461	0.478	0.41
N9	0	0.597	0.56	0.538	0.55	0.479
	5	0.59	0.54	0.522	0.537	0.472
	10	0.578	0.538	0.497	0.497	0.469
	15	0.545	0.535	0.496	0.476	0.461
	20	0.54	0.509	0.479	0.466	0.456
	25	0.515	0.501	0.464	0.465	0.41
N10	0	0.598	0.541	0.539	0.532	0.475
	5	0.566	0.54	0.505	0.531	0.473
	10	0.561	0.527	0.471	0.53	0.471
	15	0.56	0.509	0.47	0.475	0.448
	20	0.55	0.495	0.468	0.456	0.418
	25	0.546	0.487	0.466	0.451	0.414

**Table 6 entropy-24-01272-t006:** NCV-G$S^{3}$ mean and standard deviation values computed on intra-layer edges for all the alignments by applying Infomap, Generalized Louvain, ABACUS, clique percolation, and mdlp community detection algorithms.

Network	Measure	NCV-G$S^{3}$ with Infomap	NCV-G$S^{3}$ with Generalized Louvain	NCV-G$S^{3}$ with ABACUS	NCV-G$S^{3}$ with Clique Percolation	NCV-G$S^{3}$ with mdlp
N1	mean	0.945	0.927	0.928	0.921	0.929
	sd	0.035	0.047	0.028	0.042	0.027
N2	mean	0.936	0.960	0.815	0.868	0.763
	sd	0.044	0.038	0.073	0.066	0.130
N3	mean	0.955	0.947	0.875	0.813	0.784
	sd	0.026	0.040	0.079	0.077	0.073
N4	mean	0.934	0.932	0.876	0.814	0.730
	sd	0.045	0.046	0.080	0.048	0.046
N5	mean	0.951	0.948	0.873	0.858	0.756
	sd	0.030	0.033	0.051	0.067	0.063
N6	mean	0.941	0.935	0.884	0.789	0.845
	sd	0.037	0.020	0.086	0.048	0.105
N7	mean	0.943	0.945	0.875	0.795	0.884
	sd	0.030	0.041	0.047	0.045	0.109
N8	mean	0.962	0.943	0.868	0.828	0.817
	sd	0.026	0.024	0.085	0.075	0.077
N9	mean	0.921	0.948	0.895	0.894	0.772
	sd	0.043	0.031	0.089	0.079	0.140
N10	mean	0.944	0.949	0.860	0.890	0.722
	sd	0.037	0.025	0.047	0.082	0.100

**Table 7 entropy-24-01272-t007:** NCV-G$S^{3}$ mean and standard deviation values computed on inter-layer edges for all the alignments by applying Infomap, Generalized Louvain, ABACUS, clique percolation, and mdlp community detection algorithms.

Network	Measure	NCV-G$S^{3}$ with Infomap	NCV-G$S^{3}$ with Generalized Louvain	NCV-G$S^{3}$ with ABACUS	NCV-G$S^{3}$ with Clique Percolation	NCV-G$S^{3}$ with mdlp
N1	mean	0.675	0.675	0.657	0.673	0.542
	sd	0.058	0.058	0.065	0.044	0.029
N2	mean	0.697	0.681	0.693	0.637	0.543
	sd	0.041	0.071	0.049	0.063	0.037
N3	mean	0.691	0.662	0.644	0.656	0.556
	sd	0.046	0.066	0.066	0.048	0.043
N4	mean	0.642	0.635	0.622	0.651	0.539
	sd	0.021	0.083	0.025	0.059	0.038
N5	mean	0.711	0.667	0.656	0.644	0.553
	sd	0.063	0.092	0.051	0.062	0.033
N6	mean	0.748	0.703	0.616	0.657	0.542
	sd	0.056	0.053	0.052	0.083	0.031
N7	mean	0.702	0.661	0.633	0.591	0.548
	sd	0.041	0.051	0.056	0.029	0.038
N8	mean	0.682	0.649	0.620	0.654	0.534
	sd	0.066	0.070	0.068	0.041	0.030
N9	mean	0.693	0.708	0.624	0.644	0.553
	sd	0.059	0.070	0.080	0.055	0.035
N10	mean	0.707	0.708	0.660	0.655	0.556
	sd	0.054	0.058	0.053	0.068	0.035

**Table 8 entropy-24-01272-t008:** F-NC mean and standard deviation values computed on intra-layer edges for all the alignments by applying Infomap, Generalized Louvain, ABACUS, clique percolation, and mdlp community detection algorithms.

Network	Measure	F-NC with Infomap	F-NC with Generalized Louvain	F-NC with ABACUS	F-NC with Clique Percolation	F-NC with mdlp
N1	mean	0.604	0.550	0.543	0.509	0.522
	sd	0.020	0.020	0.037	0.014	0.031
N2	mean	0.618	0.584	0.551	0.534	0.492
	sd	0.022	0.031	0.018	0.017	0.036
N3	mean	0.599	0.594	0.534	0.537	0.509
	sd	0.028	0.014	0.027	0.028	0.033
N4	mean	0.593	0.564	0.543	0.551	0.523
	sd	0.032	0.036	0.038	0.021	0.028
N5	mean	0.636	0.556	0.562	0.499	0.495
	sd	0.012	0.023	0.036	0.036	0.030
N6	mean	0.594	0.546	0.539	0.536	0.493
	sd	0.033	0.027	0.036	0.033	0.025
N7	mean	0.598	0.563	0.535	0.514	0.516
	sd	0.035	0.044	0.031	0.034	0.028
N8	mean	0.584	0.547	0.553	0.501	0.515
	sd	0.029	0.026	0.019	0.036	0.029
N9	mean	0.602	0.567	0.560	0.539	0.501
	sd	0.029	0.031	0.032	0.025	0.042
N10	mean	0.586	0.550	0.529	0.512	0.479
	sd	0.038	0.034	0.033	0.032	0.016

**Table 9 entropy-24-01272-t009:** F-NC mean and standard deviation values computed on inter-layer edges for all the alignments by applying Infomap, Generalized Louvain, ABACUS, clique percolation, and mdlp community detection algorithms.

Network	Measure	F-NC with Infomap	F-NC with Generalized Louvain	F-NC with ABACUS	F-NC with Clique Percolation	F-NC with mdlp
N1	mean	0.538	0.534	0.496	0.510	0.451
	sd	0.022	0.036	0.033	0.035	0.014
N2	mean	0.534	0.528	0.513	0.490	0.447
	sd	0.037	0.025	0.033	0.029	0.037
N3	mean	0.543	0.533	0.504	0.506	0.448
	sd	0.023	0.035	0.027	0.037	0.021
N4	mean	0.546	0.540	0.509	0.500	0.444
	sd	0.030	0.040	0.026	0.030	0.023
N5	mean	0.543	0.533	0.504	0.506	0.448
	sd	0.023	0.035	0.027	0.037	0.021
N6	mean	0.558	0.508	0.509	0.490	0.423
	sd	0.037	0.033	0.024	0.030	0.029
N7	mean	0.547	0.506	0.496	0.508	0.465
	sd	0.030	0.015	0.026	0.032	0.024
N8	mean	0.564	0.549	0.509	0.494	0.443
	sd	0.030	0.035	0.034	0.018	0.031
N9	mean	0.561	0.531	0.499	0.499	0.458
	sd	0.032	0.022	0.027	0.037	0.025
N10	mean	0.564	0.517	0.487	0.496	0.450
	sd	0.018	0.023	0.030	0.039	0.028

## Data Availability

https://github.com/mmilano87/MultiLoAl (accessed on 6 September 2022).

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
