# Peer review of "Design and Implementation of a New Local Alignment Algorithm for Multilayer Networks"

_entropy, 2022, doi:10.3390/e24091272_

Round 1

Reviewer 1 Report

Review report on “Design and Implementation of new Local Alignment Algorithm for Multilayer Networks”

In the manuscript, the authors propose a novel, heuristic algorithm for the local alignment of multilayer networks, MultiLoAl (Local Alignment of Multilayer Networks). The algorithm inputs two multilayer graphs and a list of seed nodes, and outputs a set of multilayer graphs representing single local regions of similarities. The algorithm contains two key steps: 1) merging two input multilayer graphs into a single one, called multilayer alignment graph,

2) apply a community detection algorithm to the multilayer alignment graph to detect local regions of similarities. The authors present an evaluation of the algorithm by computing NCV-GS3 and F-NC values on a synthetic dataset of multilayer networks.

The paper is clearly written. To the best of my knowledge, the method is new. However, there is room for improvement concerning the method. I list some concerns as follows:

First, it seems to me that the first step of the algorithm is essentially step 1 in L-HetNetAligner,

if we consider heterogeneous networks (node-colored graphs) as multilayer networks. In the meantime, the treatments of interlayer edges and intralayer edges are different in MultiLoAl, see Fig. 9. I would suggest a comparison of the two methods MultiLoAl and L-HetNetAligner to point out their commons and differences.

Second, it is unclear how to match two nodes in the first step in general. In line 176, the authors mention that “each node is matched with the most similar node of the other network”.

I would recommend the authors provide clarification on how to obtain the similarity between two nodes in general.

Third, the description of adding inter-layer edges is unclear, see lines 202—210 and Fig. 9 a) and b). Why are heterogeneous matches and mismatches related to two nodes G1,G2 in layer 1 and two nodes D1,D2 in layer 2? It seems to me that it is determined by G1-G1 and D1-D1.

All in all, I recommend the paper be published after revision. I have included more precise suggestions for changes below.

Page 1, line 2. “Application of network alignment” -> “Applications of network alignment”

Page 1, Line 7. “Examples of such networks are the social, financial, molecular, and brain.”  the sentence reads odd.

Page 1 line 20, “comprise the mining of the single network” -> “comprise the mining of a single network”

Page 2, line 40. The notation “ EinterLxL  ” is not defined. Similarly, Einterv xk

Page 2, line 40. “L = {0, 1, ... l}” -> “L = {0, 1, ... , l}”.

Page 2, line 54. “MultiAli” -> “MultiLoAl”.

Page 2, line 57. “according similarity relationships” -> “according to similarity relationships”

Page 3, line 100. “is became a hot research topic” -> “has become a hot research topic”

Page 4, line 110. “a methods” -> “a method”

Page 4, line 150. “MultiLoAli” -> “MultiLoAl”

Resolution issue of the Figures: some labels in the figures are difficult to recognize. I recommend using vector graphics to output figures with higher resolution, such as eps format.

Table 2—5, I suggest adding a summary for each table and providing mean values and standard deviations.

Author Response

Review report on “Design and Implementation of new Local Alignment Algorithm for Multilayer Networks”

In the manuscript, the authors propose a novel, heuristic algorithm for the local alignment of multilayer networks, MultiLoAl (Local Alignment of Multilayer Networks). The algorithm inputs two multilayer graphs and a list of seed nodes, and outputs a set of multilayer graphs representing single local regions of similarities.

The algorithm contains two key steps: 1) merging two input multilayer graphs into a single one, called multilayer alignment graph,

2) apply a community detection algorithm to the multilayer alignment graph to detect local regions of similarities. The authors present an evaluation of the algorithm by computing NCV-GS3 and F-NC values on a synthetic dataset of multilayer networks.

The paper is clearly written. To the best of my knowledge, the method is new. However, there is room for improvement concerning the method. I list some concerns as follows:

First, it seems to me that the first step of the algorithm is essentially step 1 in L-HetNetAligner,

if we consider heterogeneous networks (node-colored graphs) as multilayer networks. In the meantime, the treatments of interlayer edges and intralayer edges are different in MultiLoAl, see Fig. 9. I would suggest a comparison of the two methods MultiLoAl and L-HetNetAligner to point out their commons and differences.

Answer: We apologize since we were not able to clarify this point. We added a paragraph that discusses the commons and differences among MultiLoAl and L-HetNetAligner. We added the following subsection explaining such differences:

->

\subsection{MultiLoAl vs L-HetNetAligner}

 MultiLoAl despite it is based on the previous L-HetNetAligner, it presents many different characteristics.

First,  algorithms have a different scope: MultiLoAl is a local aligner of multilayer networks while, L-HetNetAligner works only on heterogeneous networks.

In detail, by analyzing the building of local alignment, MultiLoAl and L-HetNetAligner have two main general steps: (i) construction of the  alignment graph, (ii) mining of the alignment graph.  The building of the alignment graph is the first main difference among the two algorithms.

In fact,  MultiLoAl builds a multilayer alignment graph through two substeps: (i) by adding nodes and intralayer edges, following the approach proposed in L-HetNetAligner adapted in the case of one-color networks; (ii) then, by adding interlayer edges. This last step represents the main novelty compared to L-HetNetAligner, because MultiLoAl analyzes and adds the edges among different layers of  input networks. 

Otherwise, L-HetNetAligner builds an heterogeneous alignment graph. Initially, L-HetNetAligner defines the nodes of the alignment graph as represented as composite nodes representing pairs of nodes matched by the similarity considerations. 

The algorithm inserts  and weights the edges in the alignment graph to  the nodes that the edge links have the same color and according to their distance in the input network.

Finally, once the alignment graph is built, both algorithms mine the alignment graph to discover modules that represent local alignment.

MultiLoAl applies a  community detection algorithm, Infomap, to mine the final alignment.

 The result consists of the extracted communities as a list of  nodes, the weight of the edge and the string in which is reported if there is a homogeneous/heterogeneous match, homogeneous/mismatch, homogeneous gap

Otherwise, L-HetNetAligner uses the Markov clustering (MCL) algorithm to cluster the graph. Each extracted module represents a single region of the alignment. The result of our algorithm is a list of mapping among a subset of nodes of two networks, i.e. a set of mapped regions among input graphs.

Second, it is unclear how to match two nodes in the first step in general. In line 176, the authors mention that “each node is matched with the most similar node of the other network”.

Answer:  We apologize since we were not able to clarify this point. We want to point out that similar nodes are given as input, and the algorithm selects all the similar nodes that are given as input. This procedure is commonly used in almost all the local alignment algorithms, see for instance L-HetNetAligner, Mawish, NetworkBlast, AlignNemo and Align-MCL, as pointed out in Guzzi and Milenkovic Brief Bioinfo 2019. Meng et al Bioinformatics 2016, Guzzi and Milenkovic Briefings in Bioinformatics 2017, MINA and Guzzi IEEE TCBB 2014).

In order to make these concept clearer in the paper, we also modified the text as follows:

 We rewrote the paragraph at lines 176-188-> 

At this stage, the algorithm, starting  

from an initial list of seed nodes, builds the alignment graph by initially constructing two intermediate alignment graphs, which we called  alignment graph layer 1 and alignment graph layer 2, for two networks belonging to layer 1 and two networks belonging to layer 2.

So, we define the alignment graph $G{al}_=(V_{al},E_{al})$ as a graph constructed by  two initial input graphs $G_1=(V_{1},E_{1})$, and $G_2=(V_{2},E_{2})$. Each node $v_{al}\in V_{al}$ represents a match of nodes of the input graphs, so $V_{al} \subseteq V_{1} \times V_{2}$.

The selection of node pairs is guided by the input similarity relationships. Therefore, each node is matched with the most similar node of the other network through the use of the input similarity relationships, i.e. seed nodes; and each node of the alignment graph represents a pair of similarity among nodes from the input networks, see Figure \ref{fig:diap7}. 

I would recommend the authors provide clarification on how to obtain the similarity between two nodes in general.

Answer:  We apologize since we were not able to clarify this point.  We rewrote the  the paragraph at lines 58-63 -> 

 Our algorithm receives as two multilayer networks and   a set of similarity relationships among nodes  of the same layer in both networks used as seed to build the alignment.

For instance, considering biological networks, similarity relations are represented by orthologs. The user may find these relations in databases of orthologs (e.g. OrthoMCL etc).

Third, the description of adding inter-layer edges is unclear, see lines 202—210 and Fig. 9 a) and b). Why are heterogeneous matches and mismatches related to two nodes G1,G2 in layer 1 and two nodes D1,D2 in layer 2? It seems to me that it is determined by G1-G1 and D1-D1.

Answer: We apologize since we were not able to clarify this point. We rewrote the section.

-> 

The algorithm adds the inter-layer edges among the multilayer  alignment graph layer 1 and alignment graph layer 2.

For each  pair of nodes in the multilayer  alignment graphs, the algorithm  examines the corresponding layers of the input graphs.

Let us consider the pair of nodes $(G1)$ and $(D4)$ in Figure \ref{fig:diap4}.  To determine the presence of an edge, we consider the edges $(G1,D4) \in G_1$ network and $ (G1,D4) \in G_2$ network.

The initial graph contains both the edges connecting their internal nodes, and the nodes are adjacent, there is a \textbf{match},  that we call, for convenience, \textbf{heterogeneous match}, since the nodes of the two networks are of the different type,  see Figure \ref{fig:diap8} (a).

Let us consider the pair of nodes $(G5)$ and $(D2)$ in Figure \ref{fig:diap4} (b).  To determine the presence of an edge, we consider the edges $(G5,D2) \in G_1$ network and $ (G5,D2) \in G_2$ network.

$G_1$ contains the edge $(G5,D2)$, while nodes $ G5$ and $D2$  are disconnected in $G_2$

If the initial graph contains both the edges connecting their internal nodes, and the nodes are adjacent, there is a \textbf{match},  that we call, for convenience, \textbf{heterogeneous match}, since the nodes of the two networks are of the different type,  see Figure \ref{fig:diap8} (a).

 Therefore, there is a  \textbf{heterogeneous mismatch}  Figure \ref{fig:diap8} (b).

 Then, we set the weight assigned to each edge as follow:

heterogeneous match equal to 0.9, heterogeneous mismatch equal to 0.4.

All in all, I recommend the paper be published after revision. I have included more precise suggestions for changes below.

Page 1, line 2. “Application of network alignment” -> “Applications of network alignment”

 Answer: We apologize for the issue. We fixed this.

Page 1, Line 7. “Examples of such networks are the social, financial, molecular, and brain.”  the sentence reads odd.

Answer: We apologize for the issue. We fixed this. We rewrote the sentence.

-> 

Multilayer networks are common in many application scenarios, such as modelling of relations among people in a social network, representing interplay of different molecules in a  cell or different cells in the brain.

Page 1 line 20, “comprise the mining of the single network” -> “comprise the mining of a single network”

Answer: We apologize for the issue. We fixed this.

Page 2, line 40. The notation “ EinterLxL  ” is not defined. Similarly, Einterv xk

 Answer: We apologize for the issue. We fixed this.

Page 2, line 40. “L = {0, 1, ... l}” -> “L = {0, 1, ... , l}”.

Answer: We apologize for the issue. We fixed this.

Page 2, line 54. “MultiAli” -> “MultiLoAl”.

 Answer: We apologize for the issue. We fixed this.

Page 2, line 57. “according similarity relationships” -> “according to similarity relationships”

Answer: We apologize for the issue. We fixed this.

Page 3, line 100. “is became a hot research topic” -> “has become a hot research topic”

Answer: We apologize for the issue. We fixed this.

Page 4, line 110. “a methods” -> “a method”

Answer: We apologize for the issue. We fixed this.

Page 4, line 150. “MultiLoAli” -> “MultiLoAl”

Answer: We apologize for the issue. We fixed this.

 Resolution issue of the Figures: some labels in the figures are difficult to recognize. I recommend using vector graphics to output figures with higher resolution, such as eps format.

Answer:  We thank the reviewer for pointing out this. We improved the resolution of Figures by using eps format.

Table 2—5, I suggest adding a summary for each table and providing mean values and standard deviations.

Answer:  We thank the reviewer for pointing out this. We added Tables 6-9 that report the mean and standard deviation values of NCV-G$S^3$ and F-NC for each synthetic network aligned with its noisy counterpart.

Reviewer 2 Report

1. What is the purpose of this work? It appears to be a review paper but readers can easily pick up a textbook to read up all these materials. Authors should carefully explain the rationale of this work in the main text and also orientate the readers about the different sections.

2. What is the difference between the current work and "An Extensive Assessment of Network Embedding in PPI Network Alignment" published in this same journal?

3. Section 4.2 seems to be the only original results, while the other sections read just like a thesis. 

I am unsure if this is worthy of publication in this  journal. Authors should addres the above concerns first.

Author Response

  1. What is the purpose of this work? It appears to be a review paper but readers can easily pick up a textbook to read up all these materials. 

Answer:  We are very sorry that we were unable to clearly show that our manuscript is an original work and not a survey.

Indeed, the manuscript is not a survey and, apart the Background section, the remainder of the work is original and it presents a novel algorithm for the local alignment of protein-protein interaction networks, represented as multilayer networks.

Authors should carefully explain the rationale of this work in the main text and also orientate the readers about the different sections.

Answer: We are grateful to the Reviewer for pointing out this. We explained the rationale of the work in the abstract (we added these new sentences) and we better explained the content of the various sections in the last part of Introduction.

 We added the following sentence in the introduction, clarifying the contribution of this paper.  ->

Main contributions of this paper are: (i) the implementation of  a novel algorithm for the local alignment of multilayer networks, (ii) the definition of  the local alignment of multilayer networks and (iii) the solution of heuristic for solving it, (iv) the implementation of  synthetic multilayer networks generator to build the data for the algorithm evaluation.

Also,  we explained the structure of the paper at the end of introduction as follows:

Section 2 discusses the background on multilayer networks and multilayer community detection, Section 3 presents the MultiLoAl Algorithm, and Section 4 presents and discusses the results. Finally, Section 5 concludes the paper.

  1. What is the difference between the current work and "An Extensive Assessment of Network Embedding in PPI Network Alignment" published in this same journal?

Answer: We apologize for the lack of clarity, the works are significantly different. This current work presents the implementation of  a novel algorithm for the local alignment of multilayer networks. On the other hand, the paper “An Extensive Assessment of Network Embedding in PPI Network  Alignment” is a survey about a  different topic, i.e. network embedding applied to alignment of PPI Network. In detail, in that survey, we presented an overview of current PPI network embedding alignment methods, a comparison among them, and a comparison to classical PPI network alignment algorithms.

  1. Section 4.2 seems to be the only original results, while the other sections read just like a thesis. 

Answer: We apologize for the lack of clarity, the Section 4.2 presents the results of the application of MultiLoAL for the alignment of synthetic networks. Moreover, also Section 3 is original because it presents in detail the MultiLoAL Algorithm as a novel algorithm for the alignment of multilayer networks. Only the Section 2 discusses the background on Alignment of Multilayer Networks and  Community Detection in Multilayer Networks.

Round 2

Reviewer 1 Report

Most of my previous concerns are addressed and/or clarified properly. I find the revisions adequate and have no further comments.

Reviewer 2 Report

Some minor issues before the revised manuscript can be accepted. Please proofread accordingly. For example:

The number of steps should be unified. 

Step (i,a)

Step (i,b)

Step (2)

 And

 Otherwise, 318

Table 6, Table 7, Table 8, Table ??